# Enhanced Specific Mechanism of Separation by Polymeric Membrane Modification—A Short Review

**DOI:** 10.3390/membranes11120942

**Published:** 2021-11-29

**Authors:** Anna Siekierka, Katarzyna Smolińska-Kempisty, Joanna Wolska

**Affiliations:** Department of Process Engineering and Technology of Polymer and Carbon Materials, Wroclaw University of Science and Technology, Wyb. St. Wyspiańskiego 27, 50-370 Wrocław, Poland

**Keywords:** modification of polymer membranes, mechanisms of phase separation, selective adsorption of compounds, ion exchange, controlled flow through membranes

## Abstract

Membrane technologies have found a significant application in separation processes in an exceeding range of industrial fields. The crucial part that is decided regarding the efficiency and effectivity of separation is the type of membrane. The membranes deal with separation problems, working under the various mechanisms of transportation of selected species. This review compares significant types of entrapped matter (ions, compounds, and particles) within membrane technology. The ion-exchange membranes, molecularly imprinted membranes, smart membranes, and adsorptive membranes are investigated. Here, we focus on the selective separation through the above types of membranes and detect their preparation methods. Firstly, the explanation of transportation and preparation of each type of membrane evaluated is provided. Next, the working and application phenomena are evaluated. Finally, the review discusses the membrane modification methods and briefly provides differences in the properties that occurred depending on the type of materials used and the modification protocol.

## 1. Introduction

The membrane is a phase that separates two other phases—it acts as a kind of filter that allows for the selective separation of the mixture, bypassing some components and retaining others [1,2]. Transport of particles through the membrane is achieved by applying a suitable driving force. The processes in which membranes are used can be classified according to the driving force used in the process. Typically, the driving force can be concentration difference (∆C) (e.g., in dialysis), pressure difference (∆P) (most relevant membrane processes such as, e.g., reverse osmosis, ultra- and microfiltration), temperature difference (∆T) (e.g., membrane distillation) [3], or generally defined by the potential chemical difference. The driving force can also be the difference in electric potentials (E) in processes such as electrolysis, electrodialysis, and capacitive deionisation [4,5,6].

The division of membranes can also be carried out due to the applied driving force of the process and many other points of view. They can be classified according to their origin into biological (e.g., cell membrane) or synthetic membranes. This synthetic group could be distinguished as organic (e.g., cellulose acetate, polysulfone, polyamide) and inorganic (ceramic, metal, glass, carbon) structures. During the division, one can also consider the morphology/structure of a given membrane, distinguishing here non-porous or porous, symmetrical or asymmetrical membranes, and so-called composite membranes.

Membranes have found many applications in various areas of our lives, ranging from separation processes, biomaterial production, catalytic processes [7], or even the creation of analytical devices—lab-on-chip (LOC) laboratories [8]. They are present in all types of industries, e.g., in the food [9] and pharmaceutical industries [10] or mines [11]. The growing importance of polymer membranes in medicine is noteworthy, e.g., drug delivery systems, artificial organs, or wound dressings [12]. The development of novel membrane materials is a major research thrust for academia, industry, and national laboratories because membrane performance is often challenged by fouling, low permeability, and high contaminant permeation relative to strict selectivity requirements [13]. The most commonly used membrane materials are synthetic polymers. Being competitive in performance and economics, they lead the membrane separation industry market. Many polymers are commercially available, but the choice of membrane polymer is not an easy task. A polymer must have appropriate properties, e.g., the polymer must be a suitable membrane in terms of its chain rigidity, chain interactions, stereoregularity, and polarity of its functional groups; must have a low binding affinity for separated molecules; must be compatible with the chosen membrane; and must be easily obtained at reasonable prices [4]. Polysulfone (PSU), polyethersulfone (PES), polyacrylonitrile (PAN), poly(vinylidene fluoride) (PVDF), polypropylene (PP), poly(phenylene oxide) (PPO), and polyamide (PA) are the membrane-forming materials commonly used today [13,14]. Among synthetic polymers, the class of biopolymers (e.g., cellulose and its derivatives, alginate, PCL, etc.) is gaining more importance in the production of membranes due to the use of less toxic, environmentally friendly, and renewable materials [15].

Because of the limited number of membrane-forming polymers, most of them are hydrophobic, negatively affecting the functional properties of the obtained membranes. Furthermore, the starting polymers from which membranes are obtained or already made membranes to use in specific processes must undergo modification processes. Therefore, the polymer modification process enables an improvement in the number of their physicochemical and processing properties. The modification of polymers is generally divided into chemical modification and physical modification.

The modification of membrane surfaces could help to change their properties and applications areas. Modified membranes possess unique properties that specifically and selectively transport chosen particles, ions, or solvents [16]. One of the most significant features belongs to modified membranes: selectivity and specificity. By selective transport of matter (ions, particles, and large compounds), the separation of complex solutions is becoming easier. The benefits came from the particular selective properties of the membranes that could reduce separation time and energy consumption, increase separation factors, or separate problematic feeds [17,18,19].

This review compares novel types of entrapped matter (ions, compounds, and particles) within membrane technology associated with smart transportations depending on the process conditions. The ion-exchange mechanism, molecularly imprinted membranes, smart membranes, and adsorptive membranes will be studied to explain the differences in the transport mechanisms and their selective behaviour. Here, we focus on the selective movement of separate and problematic particles through membranes and the strategy of surface modification and preparation of these distinguished membranes. 

## 2. Ion-Exchange Mechanism

### 2.1. Charectiristic of Ion-Exchange Membranes

The first specific types of membrane are ion-exchange membranes (IEM). Ion-exchange membranes (perm-selective, ion-selective) are membranes that contain the charge. The name IEM was borrowed from ion exchange resins. Furthermore, they are classified as ionic polymers, which contain ionic functional groups in their structure. These functional groups are introduced into the polymer molecule using an appropriate monomer or chemical modification that does not contain substituents. In the structure of the ion-exchange membrane, acidic groups (e.g., sulfonic groups) or base groups (e.g., ammonium groups) could be found [4,20,21]. IEMs differ from ion exchange resins in that the principle of their operation is not based on ion exchange but rather through the electrostatic exclusion of small uniform ions charged with groups of the polymer network. The quality of membranes is measured by ion transport numbers [22]. This effect is shown in Figure 1.

Cation exchange membranes (CEM) have negatively charged ionic acid groups built into their structure, e.g., sulfonic -SO_3_^−^, carboxylic -COO^−^, phosphonic -PO_3_^2−^, phosphine -PHO_2_^−^, phenyl -O^−^, arsenic -HAsO_3_^2−^, and selenium -SeO_3_^−^. They only carry cations, and anions are excluded. Their advantage over anion exchange membranes (AEM) is their better stability under more challenging conditions, for example, high temperature or pH [21,22,23]. Because of the groups present in AEM under such circumstances, the anion exchange groups can decompose by different mechanisms, depending on the conditions. Important examples of these unfavourable processes are E1 and E2 (Hofmann degradation) elimination or nucleophilic substitution [23,24,25].

Anion exchange membranes in their structure contain basic ionic groups with a positive charge, e.g., NR_3_^+^ (4-order amine), -NHR_2_^+^ (3-order amine), -NH_2_R^+^ (2-order amine), -NH_3_^+^ (1-order amine) row), phosphonium group -PR_3_^+^, and dialkyl sulfonium group -SR_2_^+^ [4,7,22,26]. Among IEMs, membranes with both types of ion exchange groups can also be distinguished. The following structures can be distinguished in this group of membranes. Amphoteric ion-exchange membranes contain randomly distributed negative and positive constant ions (ion exchange groups) in their matrix [22]. Bipolar membranes (BM) are both anion-exchange and cation-exchange membranes, forming an integral whole, or they can be formed by joining (lamination) 2 membranes: CEM and AEM—two layers. A water film approximately 2 nm thick forms between the membranes [22,27]. Mosaic ion exchange membranes are membranes whose structure consists of randomly distributed clusters of negative inactive and positive inactive groups [22,26]. As mentioned above, on-exchange membranes are chemically similar to ion-exchange resins. They are made of a polymer network with strongly bonded ionic groups (solid ions) with a positive or negative charge. Ion-active groups (charge carriers) are compensated by electroneutrality by mobile ions (counterions), having the opposite sign and participating in the ion exchange between the solutions separated by a membrane. The ion-exchange groups of the membrane prevent the intrusion of other ions with the same sign (co-ions) into the electrolyte solution by electrostatic repulsion. This effect is called Donnan foreclosure. Therefore, the concentration of co-ions in the membrane is much lower than that in the solution.

In contrast, the counterions are free to move in the membrane phase. The counterions in CEM are cations, and in AEM, anions. The exclusion of co-ions from the membrane phase is called ion selectivity. The ideal membrane only allows for the transport of counter-ions and is a barrier to co-ions. The high concentration of ions inside the membrane causes water absorption in aqueous solutions by osmosis. This effect causes it to swell. The Donnan exclusion is fulfilled at relatively low concentrations of solutions with a membrane. High concentrations (above 0.2 M) make the membrane non-selective, and the co-ions penetrate it, which leading to the transfer of whole salt molecules [4,20].

Ion exchange membranes are applied in various applications, generalised into two major categories [28]. The first category is the water-based category, which mainly includes electrodialysis [29,30], diffusion dialysis [31,32], and membrane and hybrid capacitive deionisation [33]. The second category is the energy-based category, which mainly includes reverse electrodialysis [34,35], fuel cells [36], redox flow battery [37], and electrolysis. In addition, IEMs are also used in other fields, one of the newer potential applications of ion-exchange membranes is their use as antimicrobial food packaging materials containing, e.g., silver or copper ions [38] or biomedical applications [39].

### 2.2. Preparation of Ion-Exchange Membranes

According to the way in which charge groups are connected to the matrix or their chemical structure, ion-exchange membranes can be further classified into homogenous and heterogeneous membranes. The charged groups are chemically bonded to or physically mixed with the membrane matrix [26]. Heterogeneous membranes are macroscopic clusters of ion-exchange polymers arranged in an inert, PVC, or phenolic polycondensate matrix. They are created by pressing a dry resin into a molten foil of an inert polymer or by dispersing an ion exchange resin in a polymer solution and then evaporating the solvent. These membranes are often characterised by low mechanical stability [20].

In homogeneous membranes, the ion exchange groups are evenly distributed throughout the polymer matrix. This type of membrane is produced as a result of [27,40], e.g., (i) polycondensation of functional monomers (with ionic groups) followed by cross-linking or polymerisation; (ii) introduction of ionic groups into the existing polymer network; (iii) dissolution of polymers loaded with anionic or cationic groups and then casting in the form of a film. They are characterised by high mechanical stability and low electrical resistance, and their degree of swelling depends on how a given polymer is cross-linked. Most membranes are microscopically heterogeneous structures. Complete homogeneity or macroscopic heterogeneity is rare, making such membranes unique structures. However, the most practical ion exchange membranes are relatively homogenous and composed of either hydrocarbon or fluorocarbon polymer films that host the ionic groups [26].

## 3. Molecularly Imprinted Membranes (MIM)

### 3.1. Mechanism of Entrapped Compounds

The second group of specific membranes are membranes that employ the molecular imprint for working. A fascinating group of membranes is those with a molecular imprint are called molecularly imprinted membranes (MIM) [41]. Molecularly imprinted polymers (MIPs) [42] are synthetic receptors polymerised in the presence of the target molecule—template. The most crucial factor of this technique is the selection of monomers and their number with respect to the template to ensure a stable pre-polymerisation complex. These materials have the capacity for specific molecular recognition toward the target molecule. This specific recognition of the template is due to functional groups and weak complementary interactions such as hydrogen bonds, Van der Waals forces, and ionic bonds that are present in the polymeric cavity. Due to the imprinting process, the membranes are selective towards the template used for its synthesis [43,44,45,46,47]. As a result, they can retain trace substances that are much smaller than the pores in the membrane during the filtration process (Figure 2). Such membranes can be obtained by several methods, for example, during monomer polymerisation in membrane pores or preparation of polymer composites with molecularly imprinted particles [41,48,49].

### 3.2. Preparation of Molecularly Imprinted Membranes

MIMs can be obtained in several methods, e.g., during polymerisation of monomers in membrane pores or preparation of polymers composites with molecularly imprinted particles, as well as during dry or wet phase inversion coagulation. The main problem with this method is the large number of inactive imprints located in the membrane structure. An alternative to these methods seems to be surface imprinting by grafting/coating. By treating membranes with plasma, free radicals can be generated on their surface. These radicals can be used as specific holders for the molecularly imprinted layer. Molecularly imprinted polymers, MIPs, are located on the surface of membranes. During the filtration process, the removed compound is retained on the membrane by MIPs. Through the pores of the membrane flows the already purified solution. Obtained in this way, the imprinted filters exhibited a four times higher affinity towards bisphenol A than materials without imprints, and they were not active in the sorption of phenol and its derivatives [49]. Moreover, a double imprinting process can increase the number of imprints on the surface of the membrane [50]. The first imprinted layer was prepared during a phase inversion process on SiO_2_ and activated carbon surfaces. A sol–gel polymerisation procedure was then conducted to prepare the second imprinted layer. When this method was used, it was possible to achieve a large rebinding capacity, fast adsorption kinetics, and high permselectivity coefficients [45].

### 3.3. Selective Properties

These membranes selectively recognise the imprinted particles in the model and the accurate solutions. For example, membranes imprinted toward cholesterol can absorb the cholesterol molecules in blood in the amount of 0.6 mg/dL at a pH of 8 within 30 min [51]. MIM imprinted toward polyphenols efficiently recovers these compounds from lemon, orange, and onion peel extracts [52]. Imprinted polysulfone membranes that remove polycyclic aromatic hydrocarbons from petrochemical wastewater exhibit a high affinity toward naphthalene molecules [53]. Due to their unique properties, imprinted membranes can be used, for example, in analytical chemistry, biotechnology, healthcare, environmental protection, and industrial development areas.

## 4. Smart Membranes

### 4.1. Mechanism of Phase Separation

An exciting possibility to control a material’s properties is grafting on the surface of the stimuli-responsive polymers (smart polymers). Macromolecules show a reversible change in their physical or chemical properties in response to environmental conditions such as temperature, pH, electric and magnetic fields, or ionic strength. The critical solution temperature (CST) characterises temperature-sensitive polymers. The lower critical solution temperature (LCST) is present in polymer-solvent systems. This kind of solutions have single-phase below the CST temperature, with phase separation occurring above CST. Conversely, systems have the upper critical solution temperature (UCST). These polymers change their properties along with the temperature change; the change is proportional to the intensity of the acting stimulus to the temperature difference. This effect results from a change such as hydrophobic–hydrophilic interactions [54]. In practice, polymers with a critical lower dissolution temperature are most often used. This group includes poly (N substituted acrylamides), e.g., poly (N-isopropyl acrylamide) [55] or poly (N-vinylisobutylamide) [56], as well as copolymers, e.g., poly (ethylene oxide) and poly (propylene oxide) [57]. The movement of compounds through the membrane with grafted smart polymer brushes depends on the environmental conditions in which the membrane works and the amount of grafted polymers. If the chains have been grafted into the membrane’s pores, the membrane can decrease the water permeability with a sufficiently high grafted yield. This arrangement makes it difficult to transport compounds across the membrane. By controlling the number of grafted brushes, we can control the size of the compounds that will flow through the membrane [54].

### 4.2. Surface Modification

Smart polymers grafted onto the surface of porous membranes allow for control of the porosity of the membranes, creating the so-called nanovalves. In the case of PNIPAM brushes, the membrane permeability changes with temperature. The response effect of this polymer is related to the hydrophilic–hydrophobic balance. PNIPAM consists of a hydrophobic backbone in the main chain and highly hydrophilic amide groups (-CONH_2_) linked to hydrophobic isopropyl groups. At a temperature below LCST, approximately 32 °C PNIPAM chains swell in aqueous solutions due to their solvation with water molecules and the formation of hydrogen bonds with the participation of amide groups. However, after exceeding 32 °C, hydrophobic interactions increase, hydrogen bonds are broken, and water is removed from the interchain spaces, which results in collapsing of the PNIPAM chain [54,55]. A similar effect, but dependent on the pH of the solution, can be obtained by grafting, for example, polyacrylic or polymethacrylic acid. The effect of pH-sensitive polymers is based on the protonation and deprotonation process [58]. When the units are without electric charge at a specified pH, the polymer chains are in the coil position; in this case, the membrane exhibits permeable properties. On the other hand, after changing the concentration of H^+^ cations, the groups building the units begin to dissociate, and the charge appears on the chains. The forces of Coulomb interaction between identical charges cause the chain segments to occupy the most distant position. As a result, the polymer molecule expands and becomes as large as possible, resulting in the stretching of the chains, blocking the flow through the pores of the membrane [59,60].

An exciting combination is the membranes on which mixed smart polymer brushes (MPBs) are composed [57]. For example, Figure 3 below shows the possibility of obtaining intelligent surfaces.

### 4.3. Selective Properties

Such surfaces show changes in their properties depending on the strength and intensity of the stimuli. The grafting of the copolymer P(NIPAM-co-AA) on porous membranes allows a change in the average pore size depending on both the temperature and the pH [61,62]. For example, in the paper [61], the authors observed that for membranes with brushes of AA: NIPAM 1:1 at 45 °C, the average pore size varied from 0.05 μm at pH 3.0 to 0.031 μm at pH 6.5. The same membrane at 20 °C changed the diameter of pores from 0.030 μm at pH 3.0 to 0 μm at pH 6.5. In effect, the described membranes can be used in a controlled separation. The degree of retention for *o*-bromocresol purple, in the case of brushes made of AA: NIPAM 1:1, varied from 26% to 100%, depending on external conditions [61].

Smart polymers grafted on membranes can also construct ion channels [63,64,65,66]. When a pH-sensitive poly (4-vinyl pyridine) (PVP) was used, a nanodevice was obtained that displays transport properties similar to those observed in biological pores. These properties can be controlled by manipulating the proton concentration in the surrounding environment. PVP brushes grafted inside the walls of PET membrane channels. The on-and-off mechanism was based on the manipulation of surface charges by protonation of the brushes layer. The gate was opened or closed by the environmental pH changes in the range 2 ≤ pH ≤ 5. Increasing the pH above the pKa of the pyridine moieties led to complete deprotonation of the brushes and, in effect, closed the ionic gate for ions [66].

In order for advanced ion-selective materials to be obtained, crown ethers are introduced on the membrane surfaces. These compounds can create complexes with metal cations, depending on the size of the crown ether ring and the size of the atomic radius of the metal. To design membranes that are able to selectively transport ions, one must use crown ether, which at the same time can show an affinity for a specific ion and forms not very stable complexes with it. The principle of operation of such membranes is based on the transport of ions through the pores of the membrane. If a stable complex is formed between the ether and the separated ion, these ions will be retained on the membrane, as is the case with the formation of specific ‘ion gates’ membranes [67,68,69,70,71].

However, copolymerisation of smart polymers with compounds complexing specific ions allowed for controlled and selective transport through the membrane’s pores. An exciting and significantly cheaper alternative to crown ethers is acrylic derivatives of ethylene glycol, e.g., diethylene glycol methyl methacrylic ether (DEGMEM) (Figure 4). These derivatives can create structures similar to those of ethers, specific ionophores that exhibit complexing properties. When this derivative is copolymerised with a thermosensitive polymer, e.g., PNIPAM, at temperatures above LCST, the chains are collapsed. In this situation, access to the ionophores is difficult while reducing the selectivity of the ion flow through the pores of the membrane. When the copolymer chains swell at a temperature lower than that of LCST, the ionophores complex the ions and support their selective transport through the membrane’s pores. In this way, the membrane can exhibit completely permeable or selective properties [71].

A beneficial property of smart membranes is their antifouling properties. The effectiveness of the self-cleaning membrane properties can be calculated from the absorption and desorption coefficients. This coefficient expresses the difference between the substance adsorbed on the membrane when smart polymers swell and the substance is removed from the membrane surface after changing, for example, temperature or pH and collapsing the chains [72,73,74].

In conclusion, smart polymer brushes grafted on the surface of membranes enable a controlled change of membrane properties, both permeability and selectivity. Additionally, these kinds of brushes reduce the fouling properties of the modified surfaces.

## 5. Adsorptive Membranes

### 5.1. Mechanism of Entrapment

Adsorptive membranes appear when polymers and powders with adsorption capacity are fixed in the membrane instead of added to the feed [75]. Adsorptive membranes have multifunctionality, including the adsorptive properties and matter transportation [76]. In addition, adsorptive membranes have a high rate, flux high permeability, low operating pressure, easy regeneration, and simple amplification [77]. Adsorptive membranes have an affinity for ions and molecules. Therefore, they are also called ‘affinity membranes’ [78]. Their mechanism of entrapment is based on chelating [79], complexion [80], or ion exchange [81] reactions. Hence, the adsorption membrane mechanism could highlight the chemisorption and physisorption, with different energy between accumulated matter and sorptive material [82].

The removal and entrapment of matter of adsorptive membranes is attributed to the electro-viscous effect, ion-exchange and complexation, electrostatic attraction and exclusion, size and charge exclusion, and combination of territorial binding and site binding hydrophobic binged for counterion with fixed charges on polymer chains. To understand the adsorption membranes, we have shown the removal mechanism of the adsorptive membrane, explained by a schematic representation of the electrical double layer in Figure 5.

The charge of the adsorptive membranes’ surface has appeared for various reasons, such as adsorption and ionisation. The functional groups belonging to the adsorbent accumulated in the adsorptive membrane will be charged (the adsorption mechanism is shown in the region A in Figure 5). The initial ions (ions a) combined with ion functional groups are diffused into the membrane of the feed solution due to ionisation. The ions (ions b) firstly enter the diffuse layer due to long-range electrostatic attraction (territorial binding), and counterions (ions c) may be transported in the diffuse layer due to ion pair formation or electrostatic attraction of ions b. The ions b are moved toward the Stern layer under an external driving force to form the outer spherical complexes. The ion pairs or complexes in the Stern layer can be regarded as site binding.

When the adsorptive membranes have porosity, the pores generate the convective mass transfer, or diffusion mass transfer occurs. When the pores have a large size, the convection mass transport will be dominated. The structure of the EDLs will form on the inner surface of the membrane’s pores (region B in Figure 5). The accumulation of ions in an adsorptive membrane with porosity is strongly dependent on pore size. When the pore size is smaller than the size of the ions in the feed, the ions are rejected due to the size exclusion effect. If the pore size is larger than the size of the ions, the ions (or molecules) in the diffusion layer may pass through the membrane pores under driving force. If the size of the membrane pores is moderate, the diffusion layer of the membrane pores may overlap. There is repulsion between the anion in the diffusion layer and the negative charge on the membrane surface, which results in the rejection of anions. The net has no charge on the adsorptive membrane surface (region C illustrated in Figure 5). Therefore, the cations can enter the diffuse layer and the Stern layer according to the external driving force to form the outer spherical complexes and inner spherical complexes. When this effect is used, the cations bound on the membrane surface can further diffuse and form complexes with internal adsorbents in the membrane. 

### 5.2. Surface Modification

Within the adsorptive membrane, the three main types could be distinguished. There are mixed matric adsorptive membranes, pore-filled adsorptive membranes, and surface adsorptive membranes. Hence, several methods of preparation could be delivered. Among them, the most popular are surface-coated adsorptive membrane, surface-deposited adsorptive membrane, surface-grafted adsorptive membrane, and surface-assembled adsorptive membrane (Figure 6).

The mixed matrix membranes are the most common adsorptive membranes. However, their preparation process is simple; however, not free from difficulties such as agglomeration of adsorbents. Adsorbents embedded in a polymer matrix have a lower adsorption capacity and longer adsorption equilibrium time. The following four types of surface adsorptive membranes are fabricated by coating, depositing, grafting, and assembling. Two steps usually prepare the surface-coated adsorptive membrane: the particles can first be loaded on the surface of the membrane by dipping and filtering and then be covered with the polymer layer formed by crosslinking or coating. The surface-deposited adsorptive membranes could be prepared by filtration deposition, the deposition of hydrothermal techniques, and the vapour deposition method.

On the other hand, the surface-grafted adsorptive membranes can be manufactured with the grafting method and a photo-induced post-synthetic polymerisation strategy to immobilise adsorbents on the membrane surface via a covalent link. The surface-assembled adsorptive membrane can be prepared by assembling polyelectrolytes via the electrostatic interaction. In general, the surface adsorptive membranes have a high adsorptive capacity and a short equilibrium time. However, the surface-deposited adsorptive membrane and the surface-assembled adsorptive membrane have the detachment risk of the deposition and assembly layers during application and regeneration. The surface-grafted adsorptive membranes may have a complicated process and harsh reaction conditions [75,83,84].

### 5.3. Selective Properties

One of the significant advances of adsorptive membranes is the flexible decoration of their surface by several types of adsorbents. The adequately chosen adsorbent is decided regarding the particular selective properties of AM. For AM preparation, the organic and inorganic adsorbents could be applied.

AMs have found great application in removing ions from aqueous solutions, especially heavy metal recovery. Polyethene films by radiation grafting of acrylamide had 6.2 mmol/g adsorption ability to Hg^2+^ and could reject 99% Hg^2+^ of a 200 ppm solution [84]. The grafted hyper-branched poly(amidoamine); HYPAM) in 0.45 μm polytetrafluoroethylene (PTFE) microfiltration membrane obtained a maximum adsorption capacity of 1.49 g/m^2^ to Cu^2+^, 37.4% rejection of 50 ppm Cu^2+^ solution, and pure water flux of 2556 L/m·h·bar [85]. The next example of the grafted porous membrane is the porous chitosan membranes with immobilised histidine. This membrane had the maximum adsorption capacity for Cu(II) ion of 3.0 mmol/g [86]. Furthermore, the incorporation of Cibacron blue F3GA in different polymers was used to increase the selective properties of AMs. The microporous poly(vinyl butyral) membranes immobilised Cibacron blue F3GA had adsorption capacities of 16.8 mmol/m^2^ for Zn^2+^, 7 mmol/m^2^ for Cu^2+^, 34.2 mmol/m^2^ for Pb^2+^, and 22.2 mmol/m^2^ for Cd^2+^, and could reach adsorption equilibrium in 15 min [87]. Hao et al. reported that PVDF adsorptive membranes grafted with caffeic acid had an adsorption capacity of 0.23 mmol/g for Cs^+^ and the selectivity factor of 8.46 for Cs^+^ versus Li^+^ [88].

The next series of membranes are used for fabrication the inorganic adsorbents. The mainly used adsorbents are Al_2_O_3_ [89,90], ZnO [91], TiO_2_ [92], Fe_2_O_3_ [93], manganese oxide (spinel-type) [94], montmorillonite [95], zeolites [96], Co-Fe_2_O_3_ [97], activated carbon and their modifications, nanotubes [98], and graphene oxide [99].

The MMM prepared with zeolite had the best adsorptive capacity for heavy metal ions. The PSF/zeolite MMM had adsorption capacities of 682 mg/g for Pb^2+^ and 122 mg/g for Ni^2+^, and the removal efficiencies of 91% Pb^2+^ from a 100 ppm feed and 42% Ni^2+^ from a 50 ppm feed [96]. The MMMs prepared with HMO and montmorillonite had a better adsorptive capacity to heavy metal ions. The PES/HMO MMM had an adsorptive capacity of 204 mg/g for Pb^2+^. The chitosan/montmorillonite membrane cross-linked by glutaraldehyde possessed the maximum capacity of 193 mg/g for palladium(II) at optimum pH 2 and was able to reach adsorption equilibrium in 90–100 min. [95]. The PSf/GO MMM had the maximum adsorption capacities of 79 mg/g for Pb^2+^, 75 mg/g for Cu^2+^, 68 mg/g for Cd^2+^, 17.5 mg/g for Ni^2+^, and 154 mg/g for Cr^6+^. Therefore, the PSF/GO MMM had a rejection of 90–96% to the ions mentioned above and a pure water flux of 15.42 L/m·h·bar [99].

The particular properties strongly depend on the applied adsorbent, and AMs could be found in an extensive range of selective processes to transport a particular rejection element. Furthermore, the particular properties could be associated with the sorption mechanism based on the EDLs and sorption in micro- and mesopores. Water is a significant natural resource for humans. As such, wastewater containing heavy metals is seen as a grave problem for the environment. Currently, adsorption is one of the standard methods used for both water purification and wastewater treatment. Adsorption relies on the physical and chemical interactions between heavy metal ions and adsorbents. Adsorptive membranes (AMs) have demonstrated high effectiveness in heavy metal removal from wastewater, owing to their exclusive structural properties. This article examined the applications of adsorptive membranes such as polymeric membranes (PMs), polymer–ceramic membranes (PCMs), electrospinning nanofiber membranes (ENMs), and nano-enhanced membranes (NEMs), which demonstrate high selectivity and adsorption capacity for heavy metal ions, as well as both advantages and disadvantages of each one, which are summarised and compared shortly.

Moreover, general theories for both adsorption isotherms and adsorption kinetics are described briefly to comprehend the adsorption process. This work will be valuable to readers in understanding the current applications of various AMs and their mechanisms in heavy metal ion adsorption, as well as the recycling methods in the heavy ion desorption process, being summarised and described clearly. Moreover, the influences of the morphological and chemical structures of AMs are presented and described in detail as well [96], which could be explained by the creation of specific complexation between complex conjugates and detected ions (e.g., sulphides in aromatic rings are suitable for capturing Hg^2+^). Hence, the specific transport through the adsorptive membrane could be limited by the specific surface area or the active centre and the effect of electrostatic accumulation (e.g., Langmuir interactions) visible in adsorptive membranes. Moreover, the exclusion of co-ions could play a crucial role in the transportation and receive a highly selective flux [100].

## 6. Summary

This short review presents the main mechanisms of selectively transporting ions, compounds, particles, or other membranes from aqueous solutions. Very advanced properties characterise all the materials described in this work. To our current knowledge, they will probably be used in many industries in the future [101]. IEMs can be used in desalination, wastewater treatment processes, and advanced energy storage systems. The increasing efficiency of these membranes will contribute to improving processes such as reverse electrodialysis, membrane capacitive deionisation, microbial fuel cells, ion exchange membrane bioreactors [102], and redox flow batteries (RFBs) [103]. Advanced smart membranes with properties dependent on environmental conditions offer many application possibilities. They can find application in liquid delivery systems [104], self-cleaning systems [105], and intelligent separation systems. Medicine in particular places many hopes on this type of material. Current research proves that it is possible to use materials that respond to various stimuli to target drugs more effectively [106,107]. Molecular imprint membranes are also very advanced materials. They are characterised by high selectivity and affinity toward the template.

For this reason, their importance as separation and sensory material is significant [108,109]. Furthermore, the main trends in the development of this type of material may improve electrochemical and optical sensors for applications in medicine and industry [110,111,112]. Therefore, a greater understanding of the mechanism of these materials is crucial to enable their global application in technical processes for sustainable development.

## Figures and Tables

**Figure 1 membranes-11-00942-f001:**
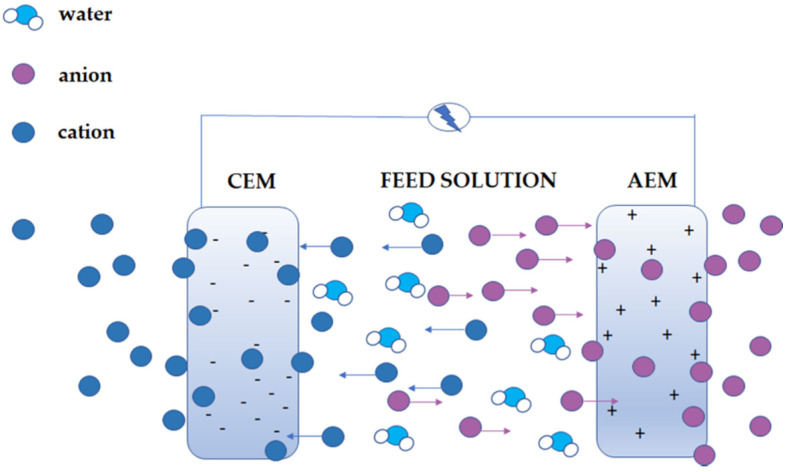
Ion-exchange membranes are used in processes such as electrodialysis that utilises an electric potential difference (ΔV) and allows passage of counter-ions (blue spheres in the case of CEM or Purple spehres in the case of AEM) while hindering co-ions (purple spheres in the case of CEM or blue spehres in the case of AEM) and water.

**Figure 2 membranes-11-00942-f002:**
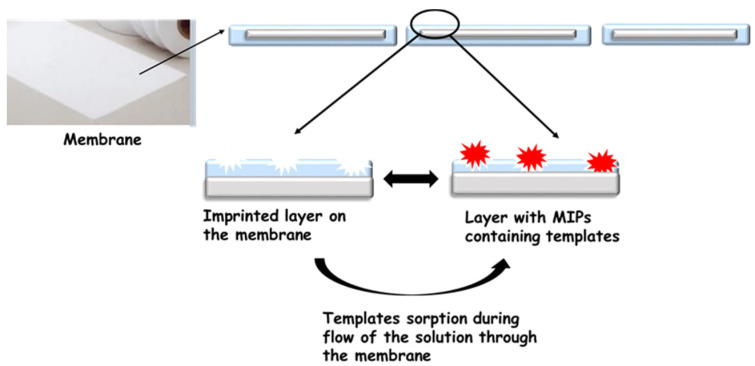
Schematic of the MIM.

**Figure 3 membranes-11-00942-f003:**
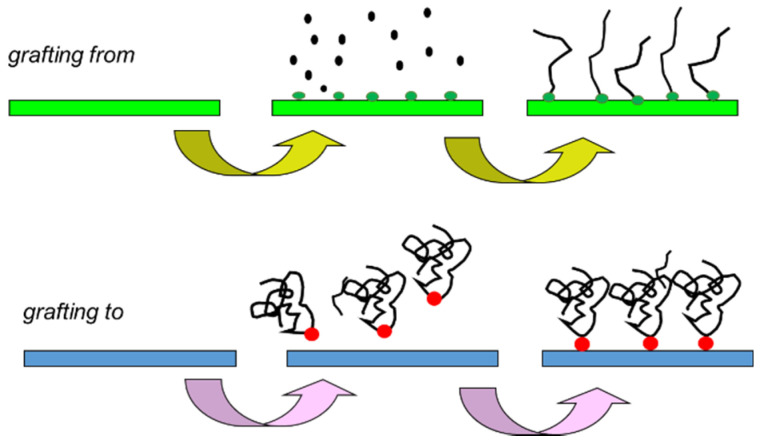
Different synthesis of smart surface possibility: “grafting to” and “grafting from”.

**Figure 4 membranes-11-00942-f004:**
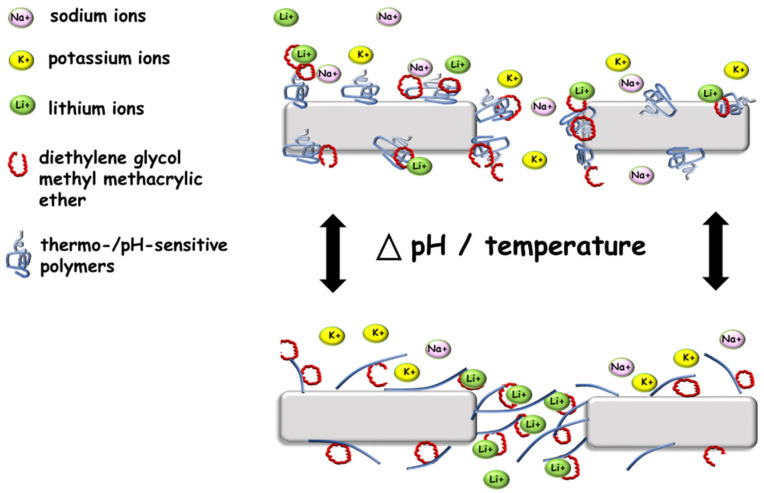
Scheme of the ion channel based on smart polymers.

**Figure 5 membranes-11-00942-f005:**
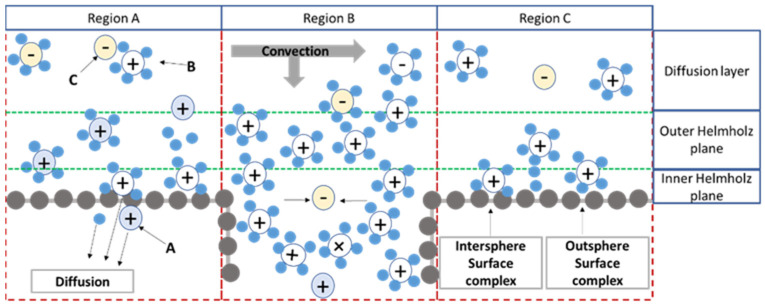
The mechanism of transportation through the adsorptive membranes. Reproduction from [75].

**Figure 6 membranes-11-00942-f006:**
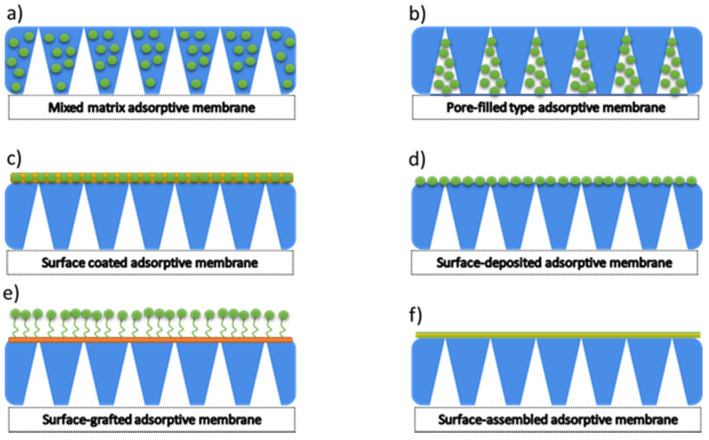
Types of adsorptive membranes. Reproduction from [75].

## Data Availability

Not applicable.

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
