# Peer review of "Enhanced Specific Mechanism of Separation by Polymeric Membrane Modification—A Short Review"

_membranes, 2021, doi:10.3390/membranes11120942_

Round 1

Reviewer 1 Report

Overall comment

The manuscript covered membrane preparation, mechanism, applications, modification and selective properties of IEM, MIM, smart membrane and adsorptive membrane. As a short review, the scope is very broad and unfortunately the listed issues were not elaborated to provide greater depth of knowledge. Meanwhile, some contents are repetitive. The manuscript must be restructured to have better continuity and to avoid repeating information. Comments for each section are as follows:

Abstract

The meaning of sentences between Line 17 and 19 are overlapping. Rephrase.

Keywords

The selected keywords did not reflect the overall content of the manuscript. Add new keywords or modify the existing.

Section 1

  • Authors must include the meaning of the term “smart-specific” under this section for clarity. The term cannot be found anywhere else other than the title.

Section 2

  • Authors should consider rephrasing Section 2.3 and merge it with 2.1 since some of the contents are overlapping. This is also for a better clarification and continuity of Section 2.
  • Page 2, Line 94. Define “primary”. Example given in the text is very general. Be specific.
  • Page 2, Line 107. Briefly elaborate the specific reasons why CEM has better stability compared to the AEM.
  • Figure 1 must be modified and improved to make it more comprehensive and able to assist the explanation under Section 2.1 and 2.3. The current figure is not easy to be understood and the details are not well presented.

Section 3

  • Page 4, Line 176. Omit the repetitive information.
  • Section 3.1, insert a figure to assist explanation on MIM.
  • Authors provided many examples on efficient MIM from literatures. The examples given were not linked to provide a concrete conclusion. In fact, Section 3.2 should focus on the mechanism of the surface modification of MIM as the title would suggest. Or else, authors should change the title to make it relevant to the content of this section.
  • Section 3.3 is repeating the information that have been provided in the earlier subsections Section 3.1 and 3.2). They are exactly the same sentences. Rewrite this section and avoid repetitive information.

Section 4

  • The title of Section 4.1 did not reflect the content. This section elaborated on the phase separation mechanism during the preparation of smart membrane instead of the movement of compound through the membrane.

Section 5

  • Redraw Figure 4. Too much information in the figure that led to confusion. It is worse because it is not self-explanatory.
  • In Section 5.3, references # [85] and [87] are irrelevant to the content of this section that was entitled surface modification. Those 2 works embedded the inorganics in the polymer matrix, hence known as a mixed matrix membrane.
  • There are many literatures cited in Section 5.3 but they are not critically reviewed. The authors failed to provide a specific conclusion on the selectivity of AMs. The last paragraph should be elaborated to provide better understanding of the selective transport in AMs.

Section 6

  • The content of Table 1 is repetitive and can be found in other subsections. Authors are advised to remove this section as the explanation should be provided in earlier subsections (Section 2 to 5).

Section 7

  • Only elaboration on IEM and MIM were found here. Provide perspectives for other types of membrane within the scope of the manuscript as well.

References

  • Very minimal recent works were cited. A review article should provide the latest information related to the field.
  • There is no information on Reference #42.

Author Response

Dear Reviewer, 
we thank for the comments and suggestions to our manuscript. We have considered the raised queries as essential for improvement quality of our work. Please find our answers in the submited file.  New changes in the manuscript are markered in yellow.
In one file there there are included respons for reviewers, the manuscript with marked changes and manuscript revised but without marked changes.

Sincerely

Katarzyna

Reviewer 2 Report

In this review paper, the authors summarized and reported on the special separation features of polymer membranes. I think this paper deals with an interesting topic, but it needs improvement in many parts to be published in "membranes". Here are my detailed comments:

1. Overall, English sentences should be reviewed and corrected by a native speaker.
2. The latest research and development trends are lacking. For example, the ion-exchange membrane section introduces just a basic overview of them.
3. A lot of careless mistakes are found in the text. Only a few of them are listed below.
- (L93) Ion -> ion
- (L94) primary -> basic
- (L95) ion exchangers -> ion-exchange resins
- (L98) ion transfer numbers -> ion transport numbers
- (L106,107) An abbreviation is defined only once when the word is used for the first time, and only the abbreviation must be used thereafter.
e.g. anion exchange membranes (AEM)
- (L118) "These" must be deleted.
- (L168,172,246,256,270,278,294,312) The reference numbers should be presented as : [30][31][32][33] -> [30-33]
- (L174) MI membranes -> MIMs
- (L205) There must be a space between the number and the unit. e.g. 0.6 mg/dL -> 0.6 mg/dL
- (L535) It is necessary to check the reference [42].
4. (L 90) "apart from ion-exchange resins and soluble polyelectrolytes," this phrase should preferably be deleted.
5. (L120-139) There is no information about surface modification of IEM.
6. (L435-453) In "Perspectives", it is considered that references should not be specified as the part where authors' opinions are written.

Author Response

Dear Reviewers, 
we thank for the comments and suggestions to our manuscript. We have considered the raised queries as essential for improvement quality of our work. Please find our answers in the submited file.  New changes in the manuscript are markered in yellow.
In one file there there are included respons for reviewers, the manuscript with marked changes and manuscript revised but without marked changes. 

Yours sincerely

Katarzyna

Round 2

Reviewer 1 Report

The authors have addressed all comments and major issues found previously have now been corrected. 

Reviewer 2 Report

The authors have carefully revised the manuscript according to the referees’ comments. In my opinion, this manuscript could be accepted for publication in membranes.